# On the Optimality Gap of
# Warm-Started Hyperparameter Optimization

**Parikshit Ram**[1]

[1]IBM Research AI

**Abstract**  We study the general framework of warm-started hyperparameter optimization (HPO) where we have some source datasets (tasks) on which we have already performed HPO, and we wish to leverage the results of these HPO to warm-start the HPO on an unseen target dataset and perform few-shot HPO. Various meta-learning schemes have been proposed over the last decade (and more) for this problem. In this paper, we theoretically analyse the optimality gap of the hyperparameter obtained via such warm-started few-shot HPO, and provide novel results for multiple existing meta-learning schemes. We show how these results allow us identify situations where certain schemes have advantage over others.

## 1 Introduction

Hyperparameter optimization or HPO has become a critical component of machine learning and artificial intelligence, where the true potential of novel learning models and techniques are only visible if the hyperparameters (HPs) are appropriately configured. However, it is also widely accepted that HPO is generally a very computationally expensive process on account of being a derivative-free optimization (DFO) problem, requiring multiple model trainings for the same problem. The (already expensive) computational cost continues to grow as the community explores larger and larger models, which take more and more compute to train (even for a single HP). There are two common techniques to mitigate this computational challenge. The first technique leverages *multi-fidelity* HPO where the cost of the model training for any HP is adaptively modified to provide more resources to more promising HPs while terminating the model training early for low performant HPs (Jamieson and Talwalkar, 2016; Sabharwal et al., 2016; Klein et al., 2017; Falkner et al., 2018). The second technique uses the "experience" from previously solved HPO problems to "warm-start" the HPO for a new problem so that we are able to achieve good performance with the evaluation of just a small number of HPs − namely, *few-shot HPO*.

In this paper, we focus on this warm-started few-shot HPO. Various such techniques have been developed under the (quite wide) umbrella of meta-learning (Vanschoren, 2018; Hospedales et al., 2021). HPO with sequential model based optimization or SMBO usually (i) performs an initial exploration of the HP space, (ii) builds a surrogate loss function and a related acquisition function that tries to mimic the true loss we want to minimize, and balances the exploration and exploitation in the HP space. This is a simplification of the SMBO process for the ease of exposition. The extent to which the surrogate loss function is informative initially relies on the level of initial exploration. There are two high-level (not mutually exclusive) meta-learning schemes that warm-start this SMBO scheme. One scheme prunes the HP space to what is considered to be promising parts of the HP space based on previous HPO experiences, so the HPO for the new problem is only performed within this pruned space. Another scheme utilizes or *transfers* the surrogate loss functions from the previous HPO experiences for a new HPO problem, and does not rely on the initial exploration of the HP space for a sufficiently informative surrogate function. A final technique complements the above techniques by making use of problem "meta-features" that allow us to estimate the similarity between the new HPO problem at hand and the previously experienced HPO problems, allowing

us to prune the HP spaces or select the transfer surrogate loss functions in an adaptive manner, focusing more on the most similar HPO problems. The gain from such meta-features rely on the ability of these meta-features to estimate the relevant similarity between different HPO tasks.

**Motivation and contributions**. While these warm-starting schemes for HPO have been widely studied empirically, we wish to understand what kind of theoretical guarantees are possible for such warm-started few-shot HPO. We believe such guarantees allow us to understand the situations where certain schemes are expected to perform better than others, and what factors affect the performance. Finally, we believe that such guarantees allow us to explain seemingly surprising empirical observations. We specifically focus on the optimality gap of warm-started few-shot HPO and make the following novel contributions:

- We present a general theoretical framework to analyse few-shot HPO with pruned HP space, and we show how this framework allows us to bound the optimality gap of various HP pruning schemes and how these guaranteees compare to each other.

- We present a general theoretical framework to analyse few-shot HPO with transfer surrogate loss functions, and we show how these guarantees compare to those for the pruning-based schemes.

**Outline**. In the following §2, we briefly discuss the existing literature on warm-starting and meta-learning. We set up notation and some preliminary assumptions and results in §3. We study the HP space pruning based warm-starting schemes in §4 and the transfer surrogate based schemes in §5. We discuss our limitations and broader impact in §6 and conclude in §7.

## 2 Related Work

Meta-learning is a wide umbrella in machine learning (see various surveys such as Vanschoren (2018); Hospedales et al. (2021)) including but not limited to areas such as multi-task learning, representation learning, transfer learning, continual learning, few-shot supervised learning, hyperparameter optimization, and also involving areas such as learning to optimize and learning loss functions. However, general HPO is one area where we are not able to leverage gradient-based optimization for the problem specific "learning"; most other areas allow for gradients to be available and even flow between the meta-level and the problem specific level. However, there are various attempts to leverage schemes from other areas in meta-learning for HPO. Volpp et al. (2019) take inspiration from the learning to optimize framework (Andrychowicz et al., 2016; Li and Malik, 2016; Chen et al., 2017) to meta-learn acquisition functions for SMBO, while Wistuba and Grabocka (2021) leverage ideas from few-shot learning (Finn et al., 2017) to meta-learn the surrogate loss function.

In the context of HPO, one set of techniques can be classified as learning initializations for the HPO (Brazdil et al., 2003; Feurer et al., 2014, 2015; Wistuba et al., 2015c). There are various techniques to generate these initializations, and they are generally selected from the set of promising HPs in previously experienced HPO problems, and possibly ranked (with additional computation) based on the aggregate performance of these selected HPs across all seen HPO tasks. Once this set of HPs has been selected and ranked, they are used to seed the HPO for new HPO problems. In the context of few-shot HPO, this amounts to just considering the HPs in this set or performing local searches around these selected HPs. This process can be seen as an *implicit form of pruning the HP space*. There are also some meta-learning schemes (Wistuba et al., 2015a; Perrone et al., 2019) that *explicitly prune* the search space based on experienced HPO problems to remove parts of the HP space where no high performing HPs lie and allow the HPO for a new problem to focus on a (possibly much smaller) HP search space. An alternate way of warm-starting the HPO is to directly utilize the surrogate loss functions from the previously experienced SMBO based HPO (Yogatama and Mann, 2014; Feurer et al., 2018b; Perrone et al., 2018; Wistuba et al., 2018; Wistuba and Grabocka, 2021). We can either maintain a single surrogate loss function that is

progressively updated as we experience multiple HPO problems, or we can just utilize the surrogate loss functions, generated independently during the previous HPO experiences, to warm-start a new HPO problem by leveraging some (possibly weighted) combination of the individual surrogate functions to make predictions for the new HPO problem. In the few-shot HPO regime we are focusing on, we assume that *it is not possible to generate an informative surrogate loss functions for the new HPO problem at hand and we have to entirely rely on the transfer surrogate loss functions*.

Meta-features from the problem often play an important role in warm-starting few-shot HPO (Vanschoren, 2010, 2018). These meta-features allow us to estimate similarity between previously experienced HPO tasks and the new task at hand. This *estimated similarity* can then be used to improve any warm-starting scheme for few-shot HPO. For initialization based schemes, these similarities can be used to rank the initializations. For transfer surrogate based warm-starting schemes, these meta-features based similarity can be used to set the weights for the different surrogate functions, guiding the new HPO problem to focus on the predictions of the surrogate models for the "nearby" previously seen HPO problems. There are various hand-crafted meta-features. However, the additional gain from these meta-features rely heavily on their ability to accurately encode the relevant notion of similarity. To this end, there are some schemes to learn such HPO specific meta-features (Wistuba et al., 2015b; Jomaa et al., 2021b,a; Rakotoarison et al., 2022).

Beyond these above schemes, there are various other meta-learning schemes that have been employed independently or in conjunction with the above schemes. Learning curve extrapolations can be meta-learned from previous HPO problems and utilized in the new HPO at hand to reject HP candidates early based on the predictions of the meta-learned extrapolator (Rijn et al., 2015; Wistuba and Pedapati, 2020). We can also meta-learn the hyperparameter importance, thereby focusing the HPO to the smaller subspace of the HP space (Hutter et al., 2014; Probst et al., 2019).

In this paper, we theoretically study a couple of specific HP pruning schemes and the transfer surrogate schemes. We will explicitly describe the schemes that we study in the following sections. While we are focusing on few-shot HPO, we also want to study the case where the number of previously seen HPO tasks is quite high. We believe that this is a reasonable situation thanks to the plethora of datasets and executions curated on repositories such as the OpenML platform (Vanschoren et al., 2013) and, to a smaller extent, the UCI repository (Dua and Graff, 2017).

## 3  Preliminaries

We begin by setting up some notation. We utilize $[N]$ for any positive integer $N$ to denote the set of indices $\{1, 2, \ldots, N\}$. For any vector $a$, we denote its $i$-th entry as $a[i]$. We use $\tau$ to denote a single HPO problem and $D$ to denote its data distribution. We reserve $t$ as a subscript to denote the index of any previously experienced HPO problem, with $\tau_t$ denoting the $t$-th HPO problem with data distribution $D_t$. We reserve $\theta$ and $\phi$ for hyperparameters (HPs) and $\Theta$ and $\Phi$ for sets of HPs.

**HPO task description.** A hyperparameter optimization or HPO *task $\tau$* corresponds to a particular data distribution $D$ from which the training set of examples and the test examples are drawn. Each example $(x, y) \sim D$ has an input domain $X$ and an output domain $Y$ with $(x, y) \in X \times Y$. We consider a $\beta$-Lipschitz loss function $\ell : Y \times Y \to \mathbb{R}_+$ where $\ell(y, y')$ quantifies the loss of predicting $y'$ for the ground truth $y$. We denote the loss of a HP $\theta$ for the data distribution $D$ as

$$L(\theta, D) := \mathbb{E}_{S \sim D^n} \mathbb{E}_{(x,y) \sim D} \ell(y, f_{\theta,S}(x)), \tag{1}$$

where $f_{\theta,S} : X \to Y$ is the model learned from training for HP $\theta$ with a training set $S$ of $n$ samples. Note that the expectation is both over the training set $S$ and the test example $(x, y)$ sampled from the data distribution. For a target HPO task $\tau$ with distribution $D$, we wish to solve the following:

$$\min_{\theta \in \Theta} L(\theta, D). \tag{2}$$

**Source tasks.** Consider $T$ source HPO tasks $\tau_t, t \in [T]$, that have been previously solved. For each task $\tau_t$ with data distribution $D_t, t \in [T]$, we denote the corresponding input domains as $X_t, t \in [T]$ and output domains as $Y_t, t \in [T]$. For each of the solved HPO tasks, we have the set of $K$ evaluated HPs $\Phi_t = \{\theta_{t,i}, i \in [K]\}$ and the surrogate loss functions $s_t : \Theta \to \mathbb{R}$. We assume that $K$ is large enough, implying that the source HPO task is solved thoroughly. For each task, we use $\phi_t$ to denote the best tried HP, that is $\phi_t = \arg\min_{i \in [K]} L(\theta_{t,i}; D_t)$. For simplicity of exposition[1], we assume that, for each task $t \in [T]$, $\phi_t$ is optimal, that is $L(\phi_t; D_t) \approx \min_{\theta \in \Theta} L(\theta; D_t)$. We do not make any assumptions on what HPO technique is used to solve the source tasks.

**Few-shot warm-started HPO for target task.** Given the experience with the $T$ source tasks, the goal of warm-started HPO for an unseen target task $\tau$ (with data distribution $D$) is to be able to warm-start the HPO so that we are able to find strong HP candidates with a small number $k \ll K$ of HP evaluations (where $K$ HPs are evaluated on each source task $\tau_t, t \in [T]$). We consider the regime where $T$ is large enough and $k$ is small enough that $k \leq T$. For the new HPO task $\tau$, we will use $\hat{\theta} \in \Theta$ to denote the solution of the few-shot HPO and $\theta^\star \in \Theta$ to denote the optimal solution. The main contributions of this paper are novel bounds on the *optimality gap* $L(\hat{\theta}, D) - L(\theta^\star, D)$ . As with the source tasks, we do not make any assumptions on the HPO technique used for the target task – in the few-shot regime, we do not have enough "shots" to obtain a meaningful surrogate function for the target task. While restrictive, this is a realistic application of warm-started HPO.[2]

**HP space and bounded change.** We use $\theta$ to denote a HP setting with $\Theta$ denoting the HP space and $\theta \in \Theta$. Assume that there are $h$ HPs. In the most general setting, we can have $h_n$ numerical, $h_o$ ordinal and $h_c$ categorical hyperparameters, with $h = h_n + h_o + h_c$. If all the HPs are numerical, $\Theta \subseteq \mathbb{R}^h$. For simplicity of exposition, we will focus on this case. However, we discuss in Appendix A how our results would extend to the case where the HPs are mixed numerical, ordinal and categorical. We make the following assumption on the dependence of the loss function on $\theta$:

*Assumption* 3.1. For a given data distribution $D$, we assume that the loss function $L(\cdot, D) : \Theta \to \mathbb{R}_+$ is $\gamma$-Lipschitz continuous. So, $\exists \gamma > 0$ such that, for any $\theta, \theta' \in \Theta$,

$$|L(\theta, D) - L(\theta', D)| \leq \gamma \cdot \|\theta - \theta'\|. \tag{3}$$

It is not very restrictive that, for any given data distribution $D$, the loss function $L(\cdot, D) : \Theta \to \mathbb{R}_+$ has a bounded Lipschitz constant. However, such an assumption seems problematic for discrete and categorical HPs. We discuss in Appendix A.2 how we can also work with a more general notion of *modulus of continuity*. In short, we only require that, for a single "unit" of change (change a categorical HP from one category to another), the loss does not change dramatically. Note that, a trivial upper bound exists for bounded losses. However, we are more interested in more non-trivial bounds. Counterexamples do exist – in a neural network, switching the activation function, a categorical HP, from linear to relu can have a dramatic effect.

**Boundedness with change in data distribution.** While we can make a boundedness assumption for a change in the HP, such an assumption is more involved to define with data distributions. We leverage the structure of problem and utilize a common notion of distance between distributions in our analysis. We specifically show the following (see proof in Appendix B.2):

---

[1]Assuming that the optimal HPs for the source tasks have been found is quite optimistic. However, we make this assumption for the ease of exposition. Instead, we can consider that each source task is $\Delta_t$-suboptimal – that is, $L(\phi_t, D_t) - \min_{\theta \in \Theta} L(\theta, D_t) \leq \Delta_t > 0$. In this case, all our results will follow in a straightforward manner with an additive factor of $\Delta_t$ in the bounds. We did not think this would add much in terms of the novelty of the analyses.

[2]A related problem is the "medium"-shot scenario where the target task gets more evaluations. In this case, the role of the HPO scheme used for the target task will play a role and we believe that our presented theoretical framework will allow one to study the conditions under which improved performance guarantees for the target task can be achieved.

**Lemma 3.1.** *For a fixed HP $\theta \in \Theta$ and any two data distributions $D, D'$, we can show that*

$$|L(\theta, D) - L(\theta, D')| \leq \beta \cdot W_1\left(P_\theta(D), P_\theta(D')\right), \tag{4}$$

*where $P_\theta(D)$ is the distribution of $(y, f_{\theta,S}(x))$ for $(x, y) \sim D$ and $S \sim D^n$ and $f_{\theta,S}$ is the model learned with training set $S$ for HP $\theta \in \Theta$ (and $P_\theta(D')$ defined correspondingly for $D'$), and $W_1$ is the 1-Wasserstein distance between two distributions.*

*Remark.* The distribution $P_\theta(D)$ is derived from $D$ as the distribution of the pair $(y, f_{\theta,S}(x))$ of the ground truth label $y$ and the prediction of a model on the corresponding test point $x$ trained on a training set $S$ with hyperparameter $\theta$, where both the training set $S$ and the test point $(x, y)$ are sampled from $D$. For example, in regression or binary classification, this $P_\theta(D)$ is a 2-dimensional distribution derived from the original data distribution $D$.

*Remark.* The use of 1-Wasserstein distance between distributions (defined in Definition B.2) comes out naturally from the proof of Lemma 3.1. While it is intuitive that the difference between the loss of two different distributions $|L(\theta, D) - L(\theta, D')|$ on the same hyperparameter $\theta$ would depend on the "domain-gap" between distributions $D$ and $D'$, our result highlights that this difference is bounded by (a scaling of) the 1-Wasserstein distance between derived distributions $P_\theta(D)$ and $P_\theta(D')$. Some novel insights here are that (i) the domain-gap is hyperparameter dependent, and (ii) the domain-gap does not require the definition of a distance between distributions of data of different sizes and dimensionality, and is defined with a (relatively) simple 1-Wasserstein distance.

**Properties of source task surrogate loss functions**. To analyse the performance of few-shot HPO with surrogate functions, we need to characterize the smoothness and quality of the per-source-task surrogate functions $s_t, t \in [T]$. The following simple but restrictive condition assumes that a surrogate loss function $s_t : \Theta \to \mathbb{R}$ universally approximates the source task loss $L(\cdot, D_t) : \Theta \to \mathbb{R}$:

*Assumption 3.2.* For each surrogate loss function $s_t, t \in [T]$, we assume that, for some small $\epsilon > 0$

$$|L(\theta; D_t) - s_t(\theta)| \leq \epsilon \forall \theta \in \Theta. \tag{5}$$

This is a very restrictive assumption but corresponds to the highest quality surrogate loss functions. We consider this here to highlight in the sequel that such high quality surrogate functions also lead to tighter bounds. Alternately, we make the following pair of weaker assumptions:

*Assumption 3.3.* We assume that, for each task $\tau_t, t \in [T]$, the corresponding surrogate loss function $s_t : \Theta \to \mathbb{R}$ is $\omega$-smooth. That is, for some small $\omega > 0$ and any $\theta, \theta' \in \Theta$:

$$|s_t(\theta) - s_t(\theta')| \leq \omega \cdot \|\theta - \theta'\|. \tag{6}$$

*Assumption 3.4.* For each source task $\tau_t, t \in [T]$ with the surrogate loss function $s_t$ and set of HPs $\Phi_t = \{\theta_{t,i}, i \in [K]\}$ tried during the source task HPO for a large $K$,

$$|s_t(\theta_{t,i}) - L(\theta_{t,i}; D_t)| \leq \epsilon \forall i \in [K]. \tag{7}$$

Assumption 3.3 assumes that the surrogate functions are Lipschitz smooth while Assumption 3.4 assumes that the surrogate loss functions have low approximation error on the HPs seen during the HPO – the set of HPs used to generate the surrogate loss function in the first place. For common nonparametric surrogate functions such as Gaussian Processes and Random Forests, this assumption is true since the error for such regressors on the training set is zero. [3]

---

[3]One aspect of our analysis is that the surrogate function (as studied) can work as is if the surrogate model is just the negative acquisition function as long as the acquisition function (AF) is sufficiently smooth (Assumption 3.3). The AF already incorporates the uncertainty and hence is handled in our analysis (albeit implicitly). Assumption 3.4 pertains to the "quality" of the surrogate function at the evaluated hyperparameters $\{\theta_{t,i}, i \in [K]\}$ (for the source task $t \in [T]$). Note that, at the evaluated hyperparameters, the uncertainty would be 0, simplifying the difference between the actual loss value and the surrogate function value (the negative AF).

**Best possible results.** Equipped with these precise technical conditions, we establish the bounds for the optimality gap from the different warm-started few-shot HPO in the following sections §4 and §5. Given the above conditions, the *best achievable universal bound* one can expect is

$$L(\hat{\theta}, D) - L(\theta^\star, D) \leq \tilde{O}\left(\max_{\theta \in \Theta} \min_{t \in [T]} W_1(P_\theta(D), P_\theta(D_t))\right). \tag{8}$$

The best possible optimality gap is related to how similar the most similar source task is on a per-HP basis. In the following, we discuss conditions under which we might achieve zero optimality gap:

*Remark.* In the simplest case, when $D \approx D_t$, we should expect no optimality gap since we have already solved the HPO task with $D_t$. However, the above bound indicates another reasonable situation where there might be zero optimality gap: If there are a set of source tasks such that $D$ is "similar" with respect to $W_1(P_\theta(D), P_\theta(D_t))$ to any one of these source tasks $\tau_t$ for any $\theta \in \Theta$ – the most similar source task can be different for different $\theta$ – then it is possible for a warm-started few-shot HPO to obtain zero optimality gap with all the information available in the experience obtained from the source tasks $\tau_t, t \in [T]$, without requiring the target distribution to match any one of the source distributions for all HPs, which is a significantly less restrictive condition.

In the ensuing presentation, we will discuss how the optimality gaps for the different warm-starting schemes compare to this best case universal bound.

## 4 Pruned Search Spaces

As discussed in §2, there are various ways of pruning the HP space $\Theta$ to a smaller subset $\bar{\Theta} \subset \Theta$, and we will specifically study three such techniques in this section. The proofs for all the theoretical results are in Appendix C. In the most general case, for a pruned HP space $\bar{\Theta} \subset \Theta$, we show that:

**Theorem 4.1.** *Given source tasks $\tau_t, t \in [T]$, and a resulting pruned search space $\bar{\Theta} \subset \Theta$, let $\hat{\theta}$ be the result of a k-shot warm-started HPO for a target HPO task $\tau$ with data distribution $D$ and let $\theta^\star$ be the optimal HP for the target task. Then, under Assumption 3.1, the optimality gap is bounded as:*

$$L(\hat{\theta}; D) - L(\theta^\star; D) \leq \min_{t \in [T]: \phi_t \in \bar{\Theta}} \left(\gamma \cdot \max_{\theta \in \bar{\Theta}} \|\theta - \phi_t\| + 2\beta \cdot \max_{\theta \in \Theta} W_1(P_\theta(D), P_\theta(D_t))\right), \tag{9}$$

*where $P_\theta(D)$ and $P_\theta(D_t)$ are as defined in Lemma 3.1, and $W_1$ is the 1-Wasserstein distance.*

To prove this result, we make use of Assumption 3.1 and Lemma 3.1. See Appendix C.2 for details. One important detail of this result is that the best possible upper bound we can achieve is

$$L(\hat{\theta}, D) - L(\theta^\star, D) \leq \tilde{O}\left(\min_{t \in [T]} \max_{\theta \in \Theta} W_1(P_\theta(D), P_\theta(D_t))\right). \tag{10}$$

Comparing this to overall best possible gap bound (8), we see that the order of the "$\min_{t \in [T]}$" and the "$\max_{\theta \in \Theta}$" is swapped, which is looser (Theorem C.3 in Appendix C.3). This implies that, with a pruned search space, it is not generally possible to get to the best achievable bound. However, it does still imply a zero optimality gap in the case where $D \approx D_t$ for some $t \in [T]$. In the following, we discuss specific HP space pruning schemes and corresponding optimality gap bounds.

**Source best HPs + local search.** The first way to prune the HP space is to only consider the best HPs for the source tasks. That is, $\bar{\Theta} := \{\phi_t, t \in [T]\}$. A generalization of this pruning would be to perform local searches around the best HPs of the source tasks. Then, for a local search radius $\delta > 0$, the pruned search space would be defined as:

$$\bar{\Theta} := \bigcup_{t \in [T]} \{\theta \in \Theta : \|\theta - \phi_t\| \leq \delta\}. \tag{11}$$

In this case, we can show the following result assuming that the few-shot budget $k > T$:

**Corollary 4.1.** *Consider the conditions and assumptions of Theorem 4.1. Then, for $k$-shot HPO with the pruned HP space defined in (11) and $k > T$, the optimality gap is bounded as*

$$L(\hat{\theta}; D) - L(\theta^\star; D) \leq \gamma\delta + 2\beta \cdot \min_{t \in [T]} \max_{\theta \in \Theta} W_1(P_\theta(D), P_\theta(D_t)). \tag{12}$$

In this case, the minimum is over all source tasks, allowing the bound to be the tightest possible as in (10). However, we would want the number of tasks $T$ to be high, in which case, $k$-shot HPO is no longer few shot. In the few-shot setting with $k < T$, we can randomly select (without replacement) $k$ out of the $T$ HPs in $\bar{\Theta}$ to seed the HPO, and have the following probabilistic bound:

**Corollary 4.2.** *Consider the conditions and assumptions of Theorem 4.1 and Corollary 4.1. Also, let us denote $\Delta_t := \max_{\theta \in \Theta} W_1(P_\theta(D), P_\theta(D_t))$ for all source task $\tau_t, t \in [T]$. Let $\Delta_{(1)} \leq \Delta_{(2)} \leq \cdots \leq \Delta_{(T)}$ be an ordering of $\{\Delta_t, t \in [T]\}$. Then, with probability at least $1 - \varepsilon$ for $\varepsilon \in (0, 1)$,*

$$L(\hat{\theta}; D) - L(\theta^\star; D) \leq \gamma\delta + 2\beta \cdot \Delta_{(n(\varepsilon))}, \text{ where } n(\varepsilon) = \min\left\{n \in [T] : \sum_{i=0}^{n-1} \frac{\binom{T-n+i}{k-1}}{\binom{T}{k}} \geq 1 - \varepsilon\right\}. \tag{13}$$

This result allows us to bound the optimality gap with high probability (in contrast to the other bounds which are deterministic) utilizing all the source tasks $\tau_t, t \in [T]$ instead of only the ones whose best HPs are selected to seed the $k$-shot HPO (with $k < T$). This result shows that we can at best get the $n(\varepsilon)$-ranked lowest bound instead of the tightest possible (as in the case in Corollary 4.1). However, note that the expected rank achievable for the random sampling scheme is $O(T/k)$ (see Theorem C.1 in Appendix C.1), so the high probability rank $n(\varepsilon)$ will only be higher.

**Ranked source best HPs + local search**. A data-driven way of ordering the per-source-task best HPs $\Phi$ is to evaluate the per-source-task best HPs $\{\phi_t, t \in [T]\}$ on all the source task distributions $D_t, t \in [T]$ and order the $\phi_t$s with respect to their aggregate loss, $\sum_{j \in [T]} L(\phi_t; D_j)$, or their aggregate rank $\sum_{j \in [T]} \rho_j\left(L(\phi_t; D_j)\right)$, where $\rho_j\left(L(\phi_t; D_j)\right)$ is the relative rank of the HP $\phi_t$ for data distribution $D_j$ in the set $\{L(\phi_1; D_j), L(\phi_2; D_j), \ldots, L(\phi_T; D_j)\}$ (Brazdil et al., 2003; Feurer et al., 2018a, 2020). We will study the aggregate loss version of the ordering. Note that this ranking process will be computationally very expensive in the regime where $T$ is large because we would have to evaluate $O(T^2)$ pairs of HPs $\{\phi_t, t \in [T]\}$ and data distributions $\{D_t, t \in [T]\}$.

**Corollary 4.3.** *Consider the conditions and assumptions of Theorem 4.1. Then, for the pruned HP space defined by picking the best $k$ HPs in $\{\phi_t, t \in [T]\}$ with respect to the metric $m_t := \sum_{j \in [T]} L(\phi_t; D_j)$ and performing a local search with a radius $\delta > 0$, the optimality gap is bounded as*

$$L(\hat{\theta}; D) - L(\theta^\star; D) \leq \gamma\delta + \beta \cdot \min_{t \in [T]}\left(K(D, D_t) + \frac{1}{T}\sum_{j \in [T]}\left(K(D, D_j) + K(D_t, D_j)\right)\right), \tag{14}$$

*where $K(D, D') := \max_{\theta \in \Theta} W_1(P_\theta(D), P_\theta(D'))$ for any pair of task data distributions $D, D'$.*

This result shows that, with $k$-shot HPO utilizing the ranked list of per-source-task best HPs, optimality gap not only depends on the distance of the target task distribution $D$ to its "closest" source task data distribution $D_{t\star} := \arg\min_{D_t} K(D, D_t)$, but also on the average distribution distance between $D$ and all the source task distributions, and the average distribution distance between $D_{t\star}$ and all the source task distributions. This implies that we can actually get a tighter bound by considering the distribution that best balances all these terms instead. It is important to note that, if the per-source-task best HPs $\Phi$ were ordered such that the $\tau_{t\star}$ (the "closest" source task) was guaranteed to be in the top-$k$ for any target task distribution $D$ (maybe by leveraging meta-features), we would be able to get the tightest possible optimality gap for the pruned HP spaces as in Corollary 4.1 in the few-shot $k < T$ regime.

**Bounding box of source best HPs**. Instead of explicitly focusing on the best HPs from the source tasks, another way to prune the HP space to consider a small subset of the HP space that contains all the per-source-task best HPs. One such subset of the HP space is the smallest bounding box in the HP space that contains $\Phi = \{\phi_t, t \in [T]\}$ and is defined as (Perrone et al., 2019)

$$\bar{\Theta} := \left\{ \theta \in \Theta : \forall i \in [h], \theta[i] \in \left[ \min_{t \in [T]} \phi_t[i], \max_{t \in [T]} \phi_t[i] \right] \right\}. \tag{15}$$

**Corollary 4.4.** *Consider the conditions and assumptions of Theorem 4.1. Then, for the pruned search space in* (15)*, the optimality gap of $k$-shot HPO is bounded by*

$$\min_{t \in [T]} \left( \gamma \cdot \sqrt{\sum_{i \in [h]} \lambda_i^2} + 2\beta \cdot \max_{\theta \in \Theta} W_1(P_\theta(D), P_\theta(D_t)) \right), \tag{16}$$

*where $\lambda_i = \max \left\{ \left( \max_{j \in [T]} \phi_j[i] - \phi_t[i] \right), \left( \phi_t[i] - \min_{j \in [T]} \phi_j[i] \right) \right\}$.*

This result indicates that we can get the desired bound with respect to the closest source distribution $D_{t^\star}$, however, we also have to pay the price of the size of the bounding box in the HP space in terms of the distance between the best HP $\phi_{t^\star}$ for the closest source task $\tau_{t^\star}$ and its furthest corner in the bounding box. Note that this furthest corner of this bounding box may not correspond to any one particular $\phi_t, t \in [T]$. For a large bounding box, we would be able to get a tighter bound by selecting the source task which balances both the terms but the bounds will still be of the same order since $\lambda_i \geq 1/2 \left( \max_{j \in [T]} \phi_j[i] - \min_{j \in [T]} \phi_j[i] \right)$ regardless of the task selected.

**Convex hull of source best HPs**. Another subset of the HP space that contains all of $\Phi = \{\phi_t, t \in [T]\}$ is the convex hull of this set (Perrone et al., 2019), giving us the following pruned HP space

$$\bar{\Theta} := \{ \theta \in \Theta : \exists w_t \in [0, 1], t \in [T], w_1 + w_2 + \cdots + w_T = 1, \theta = w_1 \phi_1 + w_2 \phi_2 + \cdots + w_T \phi_T \}. \tag{17}$$

This is a smaller subset than the bounding box in (15). We have the following result:

**Corollary 4.5.** *Consider the conditions and assumptions of Theorem 4.1. Then, for the pruned search space in* (17)*, the optimality gap of $k$-shot HPO is bounded by*

$$\min_{t \in [T]} \left( \gamma \cdot \max_{j \in [T]} \|\phi_t - \phi_j\| + 2\beta \cdot \max_{\theta \in \Theta} W_1(P_\theta(D), P_\theta(D_t)) \right). \tag{18}$$

This results shows a tighter bound relative to the bounding box (Corollary 4.4) and theoretically corroborates the motivation and the empirical observations of Perrone et al. (2019). The upper bound in Corollary 4.5 depends on the distance between actual HPs $\phi_t, \phi_j, t, j \in [T]$ as opposed to the distance to the farthest corner in the bounding box – the ratio between the former and the latter can be as low as $O(1/\sqrt{h})$ where $h$ is the number of HPs and hence can be significant. Again, if we are able to get an ordering of the similarities between the target task $\tau$ and the source tasks $\tau_t, t \in [T]$ (for example, by leveraging meta-features), the HP space can be pruned even further to focus on the part of the HP space that corresponds to the most similar tasks. This would allow us to tighten the optimality gap bound by reducing both the terms in (16) and (18), getting us closer to the best possible for pruned HP spaces (10).

*Remark.* An important implication of these results (Corollary 4.4 and 4.5) is that the search in the pruned HP space does not need to be adaptive or accurate to enjoy these bounds on the optimality gap, and potentially explains why Perrone et al. (2019) were able to show really strong performance even with random search in these pruned HP spaces.

## 5 Surrogate Functions

While pruning the HP space utilizing the per-source-task best HPs $\{\phi_t, t \in [T]\}$, we are only paying attention to the best HP from each of the source task $\tau_t$. However, we are ignoring all the remaining experience obtained from performing the full HPO on the source tasks. One way to leverage that information as well is to make use of the surrogate loss functions $s_t : \Theta \to \mathbb{R}$ learned during the source task HPO. Given these surrogate loss functions, we perform the $k$-shot HPO for the target task with data distribution $D$ using the surrogate function defined as $s(\theta) := \sum_{t \in [T]} \alpha_t(\theta) \cdot s_t(\theta)$ for any $\theta \in \Theta$ as a weighted sum of the source task surrogate functions, where the weights $\alpha_t(\theta) \in [0, 1], t \in [T], \sum_{t \in [T]} \alpha_t(\theta) = 1 \forall \theta \in \Theta$. These weights can be:

- **fixed** to a constant value such as $\alpha_t(\theta) = 1/T \forall \theta \in \Theta, \forall t \in [T]$ or to some value based on some prior knowledge regarding the similarities between the tasks (Wistuba et al., 2018),

- **adaptive** such as $\alpha_t(\theta) = 1$ if $t = \arg \max_{j \in [T]} s_j(\theta)$ and 0 otherwise, inducing the aggregated surrogate function $s(\theta) = \max_{t \in [T]} s_t(\theta)$,

- **learned** during the HPO (Yogatama and Mann, 2014; Wistuba and Grabocka, 2021).

We will focus only on fixed and adaptive (but not learned) because we are focusing on $k$-shot HPO with really small $k$ implying that we do not have enough opportunity to learn anything. We have the following result for the general transfer surrogate loss function based $k$-shot HPO:

**Theorem 5.1.** *Given source tasks $\tau_t, t \in [T]$, and their corresponding surrogate loss functions $s_t : \Theta \to \mathbb{R}$, let $\hat{\theta}$ be the result of a k-shot warm-started HPO with the surrogate function defined as $s(\theta) := \sum_{t \in [T]} \alpha_t(\theta) s_t(\theta)$ for a target HPO task $\tau$ with data distribution $D$. Let $\theta^\star$ be the optimal HP for the target task. Then, the optimality gap is bounded as:*

$$L(\hat{\theta}; D) - L(\theta^\star; D) \leq 2 \max_{\theta \in \Theta} \sum_{t \in [T]} \alpha_t(\theta) \left( \beta \cdot W_1 \left( P_\theta(D), P_\theta(D_t) \right) + |L(\theta, D_t) - s_t(\theta)| \right). \quad (19)$$

The first term in the upper bound is the *weighted* sum of the distances between the target task data distribution $D$ and all the source tasks data distribution $D_t$, while the second term depends on the ability of the per-source-task surrogate functions $s_t$ to approximate the source task loss function $L(\cdot, D_t)$. Compared to the results in §4 for pruning based warm-starting, there are two key differences: (i) Firstly, this bound involves a weighted sum over the distributional distances. While they do not directly provide a tighter bound, if the weights $\alpha_t(\theta)$ are set in a smart way (for example, by utilizing meta-features) such that similar tasks are provided higher weights, the resulting bounds would be more favorable. (ii) Secondly, and more importantly, the bounds here are $\tilde{O}(\max_{\theta \in \Theta} \sum_{t \in [T]} \alpha_t(\theta) W_1(P_\theta(D), P_\theta(D_t)))$ which could can be improved to $\tilde{O}(\max_{\theta \in \Theta} \min_{t \in [T]} W_1(P_\theta(D), P_\theta(D_t)))$ in the best case with adaptive weights, which matches (in order) the best possible discussed in (8), and hence provide an improvement over the best possible for pruned HP spaces (10). This result indicates that the increased complexity of transfer surrogate based warm-starting can provide *guaranteed improvement* over the simpler HP space pruning based warm-starting, but only with adaptive weights.

**Corollary 5.1.** *Under conditions of Theorem 5.1 and Assumption 3.2, we bound the optimality gap as*

$$L(\hat{\theta}; D) - L(\theta^\star; D) \leq 2\epsilon + 2\beta \cdot \max_{\theta \in \Theta} \sum_{t \in [T]} \alpha_t(\theta) W_1 \left( P_\theta(D), P_\theta(D_t) \right). \quad (20)$$

The restrictive Assumption 3.2 does allow us to get the tightest possible bound for the optimality gap. However, we can also provide a bound with less restrictive assumptions:

**Corollary 5.2.** *Under conditions of Theorem 5.1 and Assumptions 3.3 & 3.4, and further denoting by the set $\Phi_t = \{\theta_{t,i}, i \in [K]\}$ the HPs evaluated during the HPO of task $\tau_t$ for each $t \in [T]$, we can bound the optimality gap as*

$$L(\hat{\theta}; D) - L(\theta^\star; D) \le 2\epsilon + 2 \cdot \max_{\theta \in \Theta} \sum_{t \in [T]} \alpha_t(\theta) \left( \beta \cdot W_1 \left( P_\theta(D), P_\theta(D_t) \right) + (\gamma + \omega) \min_{i \in [K]} \| \theta - \theta_{t,i} \| \right). \quad (21)$$

One characteristic of Corollary 5.2 is that it allows us to tighten our bounds by leveraging the closest HP from the source task HPO instead of being limited to just the best HP from the source task, where the former is smaller. This allows us to maintain a smaller value for the term involving $\| \theta - \theta_{t,i} \|$ while maximizing over $\theta \in \Theta$ in the upper bound. Again, if the surrogate loss function weights $\alpha_t(\theta)$ are set in an adaptive manner, putting more weight on source task distributions most similar to the target task distribution, this bound approaches the best possible (8).

## 6 Limitations and Broader Impact Statement

**Limitations.** On a high level, one limitation of our work is that we are comparing worst case bounds on the optimality gaps, and interpreting the implications of these bounds. However, such interpretations are contingent on the tightness of such bounds. On a technical level, we are somewhat ignoring the fact that we do not really have access to the true loss $L(\theta, D)$ for some HP $\theta$ and task data distribution $D$ and instead have an estimate for a single training set sampled from this distribution. In that case, the optimality gap will also depend on the statistical properties of this estimate, and the current analyses would not directly transfer to this more general problem setup. Moreover, while we discuss the best possible optimality gap bounds in the presence of prior information that allow us to prune the HP space more aggressively or weigh the transfer surrogate loss functions adaptively, we do not precisely describe the form and the use of such information. A related limitation is that we do not explicitly focus on meta-features in our analysis. We present the first theoretical framework to study the optimality gap of warm-started few-shot HPO, and anticipate that this framework will allow us to quantify the gains from different meta-features; we will pursue this thread in future work because we believe this would be a significant contribution and of independent interest. In this paper, we only highlight situations where meta-features (learned or otherwise) **might** play a role in obtaining improved theoretical guarantees.

**Broader impact.** We focus on establishing theoretical guarantees for algorithms that have been previously empirically and practically studied and applied to various situations. To this end, we do not anticipate this work to have significant additional impact.

## 7 Conclusion and Future Work

In this paper, we focus on warm-started few-shot HPO within the SMBO framework and rigorously analyse the optimality gap of various such warm-starting schemes, providing intuitive guarantees, and identifying situations where one scheme (transfer surrogate loss functions) perform favourably compared to another (HP space pruning). As future work, we wish to extend such analysis to other warm-starting schemes such as hyperparameter importance and learning curve extrapolation based schemes. Moreover, given the critical nature of multi-fidelity optimization for practical HPO, we also plan to extend this analysis to incorporate the multi-fidelity nature of the few-shot HPO, precisely characterising the optimality gap in terms of the optimality-efficiency tradeoff.

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
