# A Handling Mixed Numerical, Ordinal and Categorical Hyperparameters

Assuming we have $h$ HPs, if $\Theta \subset \mathbb{R}^h$, then there are various distances available such as $\|\theta - \theta'\|_\rho$ (the $\rho$-norm). The more general case is where we have $h_r$ numerical HPs, $h_o$ ordinal HPs, and $h_c$ categorical HPs; $h = h_n + h_o + h_c$. In that case, $\Theta \subset \mathbb{R}^{h_n} \times \mathbb{O}^{h_o} \times \mathbb{C}^{h_c}$, and any $\theta = (\theta_\mathbb{R}, \theta_\mathbb{O}, \theta_\mathbb{C}) \in \Theta_\mathbb{R} \times \Theta_\mathbb{O} \times \Theta_\mathbb{C}$, where $\theta_\mathbb{R} \in \Theta_\mathbb{R}, \theta_\mathbb{O} \in \Theta_\mathbb{O}, \theta_\mathbb{C} \in \Theta_\mathbb{C}$ respectively denote the continuous, integer and categorical HPs in $\theta$.

## A.1 Distance in $\Theta$

Here we will define a distance metric $d : \Theta \times \Theta \to \mathbb{R}_+$. Distances over $\mathbb{R}^{h_n} \times \mathbb{O}^{h_o}$ is available, such as $\rho$-norm. Let $d_{\mathbb{R},\mathbb{O}} : (\Theta_\mathbb{R} \times \Theta_\mathbb{O}) \times (\Theta_\mathbb{R} \times \Theta_\mathbb{O}) \to \mathbb{R}_+$ be some such distance. To define distances over categorical spaces, there are some techniques such as one described by Oh et al. (2019). Assume that each of the $C$ HPs $\theta_{\mathbb{C},k}, k \in [C]$ have $n_k$ categories $\{c_{k1}, c_{k2}, \ldots, c_{kn_k}\}$. Then we can essentially define a distance between two $\theta_\mathbb{C}, \theta'_\mathbb{C} \in \Theta_\mathbb{C}$ as:

$$d_\mathbb{C} : \Theta_\mathbb{C} \times \Theta_\mathbb{C} \to \mathbb{R}_+, \quad d_\mathbb{C}(\theta_\mathbb{C}, \theta'_\mathbb{C}) = \sum_{i=1}^{C} \mathbb{I}(\theta_\mathbb{C}[i] \neq \theta'_\mathbb{C}[i]). \tag{22}$$

**Definition A.1.** We can define a distance $d : (\Theta_\mathbb{R} \times \Theta_\mathbb{O} \times \Theta_\mathbb{C}) \times (\Theta_\mathbb{R} \times \Theta_\mathbb{O} \times \Theta_\mathbb{C}) \to \mathbb{R}_+$ between two HPs $\theta, \theta' \in \Theta$ as

$$d(\theta, \theta') = d_{\mathbb{R},\mathbb{O}}((\theta_\mathbb{R}, \theta_\mathbb{O}), (\theta'_\mathbb{R}, \theta'_\mathbb{O})) + d_\mathbb{C}(\theta_\mathbb{C}, \theta'_\mathbb{C}). \tag{23}$$

*Proposition* A.1. Given distance metrics $d_{\mathbb{R},\mathbb{O}}$ and $d_\mathbb{C}$, the function $d : \Theta \times \Theta$ defined in (23) is a valid distance metric.

## A.2 Continuity in the space of HPs $\Theta$

As stated in Assumptions 3.1 & 3.3, we assume Lipschitz continuity of the loss function $L(\theta, D)$ and the surrogate loss function $s_t(\theta), i \in [p]$ with respect to a continuous $\theta \in \Theta \subset \mathbb{R}^h$, giving us:

$$|L(\theta, D) - L(\theta', D)| \leq \gamma \cdot \|\theta - \theta'\|, \tag{24}$$
$$|s_t(\theta) - s_t(\theta')| \leq \omega \cdot \|\theta - \theta'\|. \tag{25}$$

For a more general handling of a HP space $\Theta$ consisting of mixed numerical, ordinal and categorical HPs, we can leverage the notion of "modulus of continuity". Specifically, we can assume the following:

*Assumption* A.1. Consider the following increasing concave real-valued functions $\mu_L, \mu_s : \mathbb{R}_+ \to \mathbb{R}_+$ with $\lim_{v \to 0} \mu_L(v) = \mu_L(0) = 0$ and $\lim_{v \to 0} \mu_s(v) = \mu_s(0) = 0$. Then we can say that the loss function $L(\theta, D)$ and the surrogate loss function $s_t(\theta)$ admit $\mu_L$ and $\mu_s$ as a moduli of continuity respectively with respect to the distance metric $d : \Theta \times \Theta \to \mathbb{R}_+$ defined in Definion A.1, implying that

$$|L(\theta, D) - L(\theta', D)| \leq \mu_L(d(\theta, \theta')), \tag{26}$$
$$|s_t(\theta) - s_t(\theta')| \leq \mu_s(d(\theta, \theta')). \tag{27}$$

Since we assume that $\mu_L, \mu_s$ are concave, we can say that these functions are sublinear as follows:

$$\exists \gamma_1, \gamma_2 \geq 0 : \mu_L(v) \leq \gamma_1 v + \gamma_2 \forall v \in \mathbb{R}_+, \tag{28}$$
$$\exists \omega_1, \omega_2 \geq 0 : \mu_s(v) \leq \omega_1 v + \omega_2 \forall v \in \mathbb{R}_+. \tag{29}$$

These conditions indirectly give us something similar in spirit to the guarantees of Lipschitz continuity, but is a more rigorous way of achieving such guarantees.

## B  Distances between distributions

In this section, we first discuss a general notion of distance between distributions in §B.1, and then leverage a more problem-specific notion of distance between distributions in §B.2 to bound the change in the loss function $L(\theta, D)$ with change in the distribution $D$ to $D'$.

### B.1  Variation of information distance

Here we describe the *variation of information* distance between distributions.

**Definition B.1.** For any distributions $D, D'$, let $\mathcal{J}(D, D')$ be the set of all joint distributions such that $D$ and $D'$ are their respective marginals. Let $H(\cdot)$ be the entropy of any distribution. Then the variation of information distance $U(D, D')$ between $D$ and $D'$ is defined as

$$U(D, D') = V(D, D') + V(D', D), \tag{30}$$

where $V(D, D')$ and $V(D', D)$ are defined as follows:

$$V(D, D') = \min_{J \in \mathcal{J}(D, D')} H(J) - H(D) \tag{31}$$

$$V(D', D) = \min_{J \in \mathcal{J}(D, D')} H(J) - H(D'). \tag{32}$$

However, for our problem, we do not require such a general notion.

### B.2  Proof of Lemma 3.1

For our purposes, we make use of the 1-Wasserstein distance defined as follows:

**Definition B.2.** For any pair of distributions $P, Q$ over some domain $\mathcal{X}$, let $\mathcal{J}(P, Q)$ be the set of all joint distributions over $\mathcal{X} \times \mathcal{X}$ such that $P$ and $Q$ are the marginals for any joint distribution $J \in \mathcal{J}(P, Q)$. Then the 1-Wasserstein distance is defined as:

$$W_1(P, Q) := \sup_{(x, x') \sim J : J \in \mathcal{J}(P, Q)} \|x - x'\|. \tag{33}$$

**Lemma B.1** (Lemma 3.1). *For a fixed HP $\theta \in \Theta$ and any two data distributions $D, D'$, we can show that*

$$|L(\theta, D) - L(\theta, D')| \leq \beta \cdot W_1\left(P_\theta(D), P_\theta(D')\right), \tag{34}$$

*where $P_\theta(D)$ is the distribution of $(y, f_{\theta,S}(x))$ for $(x, y) \sim D$ and $S \sim D^n$ and $f_{\theta,S}$ is the model learned with training set $S$ for HP $\theta \in \Theta$ ($P_\theta(D')$ defined correspondingly), and $W_1$ is the 1-Wasserstein distance.*

*Proof of Lemma 3.1.* By definition of $L(\theta, D)$ in (1), we have a following set of relations:

$$|L(\theta, D) - L(\theta, D')| = \left| \mathbb{E}_{S \sim D^n} \mathbb{E}_{(x,y) \sim D} \ell(y, f_{\theta,S}(x)) - \mathbb{E}_{S' \sim D'^{n'}} \mathbb{E}_{(x',y') \sim D'} \ell(y', f'_{\theta,S'}(x')) \right| \tag{35}$$

$$= \left| \mathbb{E}_{S \sim D^n} \mathbb{E}_{(x,y) \sim D} \mathbb{E}_{S' \sim D'^{n'}} \mathbb{E}_{(x',y') \sim D'} \left( \ell(y, f_{\theta,S}(x)) - \ell(y', f'_{\theta,S'}(x')) \right) \right| \tag{36}$$

$$\leq \mathbb{E}_{S \sim D^n} \mathbb{E}_{(x,y) \sim D} \mathbb{E}_{S' \sim D'^{n'}} \mathbb{E}_{(x',y') \sim D'} \beta \cdot \left\| (y, f_{\theta,S}(x)) - (y', f'_{\theta,S'}(x')) \right\| \tag{37}$$

$$\leq \beta \cdot \mathbb{E}_{S \sim D^n} \mathbb{E}_{(x,y) \sim D} \mathbb{E}_{S' \sim D'^{n'}} \mathbb{E}_{(x',y') \sim D'} \left\| (y, f_{\theta,S}(x)) - (y', f'_{\theta,S'}(x')) \right\|, \tag{38}$$

where the first pairs of equalities are by definition and the inequality is obtained via the Lipschitz continuity of the pointwise loss function $\ell : Y \times Y \rightarrow \mathbb{R}_+$.

Given the definition of the distribution $P_\theta(D)$ and $P_\theta(D')$ in the statement of Lemma 3.1, we can write the final inequality above as:

$$|L(\theta, D) - L(\theta, D')| \leq \beta \cdot \mathbb{E}_{(z_1, z_2) \sim P_\theta(D)} \mathbb{E}_{(z'_1, z'_2) \sim P_\theta(D')} \|(z_1, z_2) - (z'_1, z'_2)\| \tag{39}$$

$$\leq \beta \cdot \sup_{(z_1, z_2), (z'_1, z'_2) \sim J : J \in \mathcal{J}(P_\theta(D), P_\theta(D'))} \|(z_1, z_2) - (z'_1, z'_2)\|, \tag{40}$$

where $\mathcal{J}(P_\theta(D), P_\theta(D'))$ is the set of all joint distributions $J$ with $P_\theta(D)$ and $P_\theta(D')$ as their marginals. By Definition B.2, the above gives us the following result in the statement of the lemma:

$$|L(\theta, D) - L(\theta, D')| \leq \beta \cdot W_1(P_\theta(D), P_\theta(D')). \tag{41}$$

$\square$

## C   Technical results for §4

### C.1   Expected rank when randomly sampling $k \ll T$ HP candidates

**Theorem C.1.** *The expected rank when randomly selecting without replacement $k \ll T$ out of $T$ HP candidates, the expected rank of the best in $k$ is lower bounded by $O(T/k)$.*

*Proof.* Let $K$ denote the indices of the $k$ samples $\{t_i, i \in [k]\}$. Then the expected rank can be written as

$$\mathbb{E}(\text{rank}) = \sum_{n=1}^{T-k+1} n \cdot \Pr(n\text{-th rank} \in K, m\text{-th rank} \notin K, m = 1, \ldots, n-1) \tag{42}$$

$$= \sum_{n=1}^{T-k+1} n \cdot \frac{k}{T} \prod_{j=1}^{n-1} \frac{T-k-j}{T-j} = \frac{k}{T} \sum_{n=1}^{T-k+1} n \cdot \prod_{j=1}^{n-1} \frac{T-k-j}{T-j} \tag{43}$$

$$= \frac{k}{T} \left( 1 + \sum_{n=2}^{T-k+1} n \cdot \prod_{j=1}^{n-1} \frac{T-k-j}{T-j} \right) \tag{44}$$

$$\geq \frac{k}{T} \left( 1 + \underbrace{\sum_{n=2}^{T-k+1} n \left( \frac{T-k}{T-1} \right)^{n-1}}_{B} \right) \tag{45}$$

If we set $v = T-k/T-1$, the term $B$ above is a arithmetic-geometric series and can be evaluated according as:

$$B = 1 + \sum_{n=2}^{T-k+1} n v^{n-1} \tag{46}$$

$$= \frac{T-1}{k^2} \left( T - 1 + v^{T-k} \left( (k-1)T - k^2 + 1 \right) \right) \tag{47}$$

$$> \frac{(T-1)^2}{k^2}, \tag{48}$$

where we use the fact that $1 - v = k/T-1$, $v > 0$ and that $k \ll T$ so $(k-1)T - k^2 > 0$. Hence the expected rank is lower bounded by

$$\mathbb{E}(\text{rank}) \geq \frac{k}{T} \cdot B \tag{49}$$

$$> \frac{k}{T} \cdot \frac{(T-1)^2}{k^2} = \frac{(T-1)^2}{kT} \tag{50}$$

$$\sim O\left(\frac{T}{k}\right) \tag{51}$$

This gives us the statement of the theorem. $\qquad\square$

## C.2 Proof of Theorem 4.1

**Theorem C.2** (Theorem 4.1). *Given source tasks $\tau_t, t \in [T]$, and a resulting pruned search space $\bar{\Theta} \subset \Theta$, let $\hat{\theta}$ be the result of a $k$-shot warm-started HPO for a target HPO task $\tau$ with data distribution $D$ and let $\theta^\star$ be the optimal HP for the target task. Then, under Assumption 3.1, the optimality gap is bounded as:*

$$L(\hat{\theta}; D) - L(\theta^\star; D) \leq \min_{t \in [T]: \phi_t \in \bar{\Theta}} \left( \gamma \cdot \max_{\theta \in \bar{\Theta}} \|\theta - \phi_t\| + 2\beta \cdot \max_{\theta \in \Theta} W_1 \left( P_\theta(D), P_\theta(D_t) \right) \right), \tag{52}$$

*where $P_\theta(D)$ and $P_\theta(D_t)$ are as defined in Lemma 3.1, and $W_1$ is the 1-Wasserstein distance.*

*Proof.* Let $\phi_t$ be some source-task-best HP present in the pruned HP space $\bar{\Theta}$, and let $D_t$ be the data distribution of that source task $\tau_t$ with $\phi_t = \arg\min_{\theta \in \Theta} L(\theta, D_t)$. Then we have the following by leveraging Assumption 3.1 and Lemma 3.1:

$$L(\hat{\theta}, D) - L(\theta^\star, D) = \underbrace{L(\hat{\theta}, D) - L(\phi_t, D)}_{\leq \gamma \|\theta - \phi_t\| \text{ (by Assumption 3.1)}} + L(\phi_t, D) - L(\phi_t, D_t)$$

$$+ \underbrace{L(\phi_t, D_t) - L(\theta^\star, D_t)}_{\leq 0 \text{ by def.}} + L(\theta^\star, D_t) - L(\theta^\star, D) \tag{53}$$

$$\leq \gamma \|\theta - \phi_t\| + \beta \cdot W_1(P_{\phi_t}(D), P_{\phi_t}(D_t)) + \beta \cdot W_1(P_{\theta^\star}(D), P_{\theta^\star}(D_t)) \tag{54}$$

$$\leq \gamma \max_{\theta \in \bar{\Theta}} \|\theta - \phi_t\| + 2\beta \cdot \max_{\theta \in \Theta} W_1(P_\theta(D), P_\theta(D_t)). \tag{55}$$

Since the above is true for any $\phi_t \in \bar{\Theta}$, we can choose $t$ such that $\phi_t \in \bar{\Theta}$ that minimizes the upper bound, giving us the tightest bound and the statement of the theorem. $\qquad\square$

## C.3 Comparing best case bounds (8) vs (10)

**Theorem C.3.** *For $P_\theta(D)$ and $P_\theta(D_t)$ defined as in Lemma 3.1,*

$$\max_{\theta \in \Theta} \min_{t \in [T]} W_1 \left( P_\theta(D), P_\theta(D_t) \right) \leq \min_{t \in [T]} \max_{\theta \in \Theta} W_1 \left( P_\theta(D), P_\theta(D_t) \right). \tag{56}$$

*Proof.* Let us define the following quantities:

$$\theta^* := \arg\max_{\theta \in \Theta} \left( \min_{t \in [T]} W_1 \left( P_\theta(D), P_\theta(D_t) \right) \right), \tag{57}$$

$$t(\theta^*) := \arg\min_{t \in [T]} W_1 \left( P_{\theta^*}(D), P_{\theta^*}(D_t) \right), \tag{58}$$

$$t^* := \arg\min_{t \in [T]} \left( \max_{\theta \in \Theta} W_1 \left( P_\theta(D), P_\theta(D_t) \right) \right), \tag{59}$$

$$\theta(t^*) := \arg\max_{\theta \in \Theta} W_1 \left( P_\theta(D), P_\theta(D_{t^*}) \right). \tag{60}$$

Then we have the following by defition:

$$\max_{\theta \in \Theta} \left( \min_{t \in [T]} W_1 \left( P_\theta(D), P_\theta(D_t) \right) \right) = W_1 (P_{\theta^*}(D), P_{\theta^*}(D_{t(\theta^*)}))$$

$$\leq W_1 (P_{\theta^*}(D), P_{\theta^*}(D_{t^*}))$$

$$\leq W_1 (P_{\theta(t^*)}(D), P_{\theta(t^*)}(D_{t^*})) = \min_{t \in [T]} \left( \max_{\theta \in \Theta} W_1 \left( P_\theta(D), P_\theta(D_t) \right) \right).$$

This gives us the statement of the theorem. $\qquad \square$

## C.4 Proof of Corollary 4.1

**Corollary C.1** (Corollary 4.1). *Consider the conditions and assumptions of Theorem 4.1. Then, for $k$-shot HPO with the pruned HP space defined in* (11) *and $k > T$, the optimality gap is bounded as*

$$L(\hat{\theta}; D) - L(\theta^\star; D) \leq \gamma\delta + 2\beta \cdot \min_{t \in [T]} \max_{\theta \in \Theta} W_1(P_\theta(D), P_\theta(D_t)). \tag{61}$$

*Proof.* Given the fact that $k > T$, we know that $\phi_t \in \bar{\Theta}$ for all $t \in [T]$ with the pruned HP space $\bar{\Theta}$ defined in (11). Moreover, $\|\theta - \phi_t\| \leq \delta$ for all $t \in [T]$. Then, leveraging (9) in Theorem 4.1, we get the statement of the corollary by being able to minimize over all $t \in [T]$. $\qquad \square$

## C.5 Proof of Corollary 4.2

**Theorem C.4** (Sedransk and Meyer (1978)). *For a population of size $R$ with values $\{v_1, \dots, v_R\}$ ordered as $v_{(1)} \leq v_{(2)} \leq \dots \leq v_{(R)}$, let $w_{(1)} \leq w_{(2)} \leq \dots \leq w_{(r)}$ be an ordered sample of size $r < R$ randomly drawn from the population uniformly without replacement. Then, for $1 \leq A \leq R$ and $1 \leq a \leq r$*

$$\Pr \left( w_{(a)} \leq W_{(A)} \right) = \sum_{i=1}^{A-a} \binom{A - i - 1}{a - 1} \binom{R - A + i}{r - a} \bigg/ \binom{R}{r}. \tag{62}$$

**Corollary C.2** (Corollary 4.2). *Consider the conditions and assumptions of Theorem 4.1 and Corollary 4.1. Also, let us denote $\Delta_t := \max_{\theta \in \Theta} W_1(P_\theta(D), P_\theta(D_t))$ for all source task $\tau_t, t \in [T]$. Let $\Delta_{(1)} \leq \Delta_{(2)} \leq \dots \leq \Delta_{(T)}$ be an ordering of $\{\Delta_t, t \in [T]\}$. Then, with probability at least $1 - \varepsilon$ for $\varepsilon \in (0, 1)$,*

$$L(\hat{\theta}; D) - L(\theta^\star; D) \leq \gamma\delta + 2\beta \cdot \Delta_{(n(\varepsilon))}, \text{ where } n(\varepsilon) = \min \left\{ n \in [T] : \sum_{i=0}^{n-1} \binom{T-n+i}{k-1} \bigg/ \binom{T}{k} \geq 1 - \varepsilon \right\}. \tag{63}$$

*Proof of Corollary 4.2.* For our purposes, we are sampling $k$ HPs from a population of size $T$, so mapping it to the statement of Theorem C.4, $r = k$ and $R = T$. Let the values $\{\Delta_t, t \in [T]\}$ of the whole population be ordered as $\Delta_{(1)} \leq \Delta_{(2)} \leq \dots \leq \Delta_{(T)}$. Let $\{t_i, i \in [k]\}$ denote the indices of the $k$ HPs random sampled (without replacement) from the $T$ per-source-task HPs. Let the values $\{\Delta_{t_i}, i \in [k]\}$ be ordered as $\bar{\Delta}_{(1)} \leq \bar{\Delta}_{(2)} \leq \dots \leq \bar{\Delta}_{(k)}$.

So we want to find the smallest rank $n \leq T$ such that the probability $\Pr(\bar{\Delta}_{(1)} \leq \Delta_{(n)})$ will at least $1 - \varepsilon$. Hence we want

$$\Pr \left( \bar{\Delta}_{(1)} \leq \Delta_{(n)} \right) = \sum_{i=0}^{n-1} \binom{T - n + i}{k - 1} \bigg/ \binom{T}{k} \geq 1 - \varepsilon, \tag{64}$$

where we get the first equality by applying (62) in Theorem C.4. Hence we choose as $n(\varepsilon)$ the smallest $n$ such that $\Pr(\bar{\Delta}_{(1)} \leq \Delta_{(n)})$ will at least $1 - \varepsilon$ as in the definition of the Corollary.

Moreover, $\|\theta - \phi_{t_i}\| \leq \delta$ for all $\{t_i, i \in [k]\}$ by the definition of $\bar{\Theta}$ in (11) and the fact that $\phi_{t_i} \in \bar{\Theta} \forall i \in [k]$. Then, leveraging (9) in Theorem 4.1, we get the statement of the corollary. $\qquad \square$

## C.6 Proof of Corollary 4.3

**Corollary C.3** (Corollary 4.3). *Consider the conditions and assumptions of Theorem 4.1. Then, for the pruned HP space defined by picking the best $k$ HPs in $\{\phi_t, t \in [T]\}$ with respect to the metric $m_t := \sum_{j \in [T]} L(\phi_t; D_j)$ and performing a local search with a radius $\delta > 0$, the optimality gap is bounded as*

$$L(\hat{\theta}; D) - L(\theta^\star; D) \leq \gamma \delta + \beta \cdot \min_{t \in [T]} \left( K(D, D_t) + \frac{1}{T} \sum_{j \in [T]} \left( K(D, D_j) + K(D_t, D_j) \right) \right), \tag{65}$$

*where $K(D, D') := \max_{\theta \in \Theta} W_1(P_\theta(D), P_\theta(D'))$ for any pair of task data distributions $D, D'$.*

*Proof.* Let $\phi_t$ be some source-task-best HP present in the pruned HP space $\bar{\Theta}$, and let $D_t$ be the data distribution of that source task $\tau_t$ with $\phi_t = \arg\min_{\theta \in \Theta} L(\theta, D_t)$. Let $\phi_{t'} \in \Phi = \{\phi_t, t \in [T]\}$, where $\phi_{t'} \notin \bar{\Theta}$ and $D_{t'}$ be the corresponding source task data distribution. This implies that in the ranked list of HPs, $\sum_{j \in [T]} L(\phi_t, D_j) \leq \sum_{j \in [T]} L(\phi_{t'}, D_j)$ for any $\phi_t \in \bar{\Theta}, \phi_{t'} \notin \bar{\Theta}$.

Then we have the following:

$$L(\hat{\theta}, D) - L(\theta^\star, D) = L(\hat{\theta}, D) - L(\phi_t, D) + L(\phi_t, D) - \frac{1}{T} \sum_{j \in [T]} L(\phi_t, D_j)$$

$$\underbrace{+ \frac{1}{T} \sum_{j \in [T]} L(\phi_t, D_j) - \frac{1}{T} \sum_{j \in [T]} L(\phi_{t'}, D_j)}_{\leq 0 (\text{see discussion above})}$$

$$+ \frac{1}{T} \sum_{j \in [T]} L(\phi_{t'}, D_j) - \frac{1}{T} \sum_{j \in [T]} L(\phi_{t'}, D_{t'})$$

$$+ L(\phi_{t'}, D_{t'}) - L(\theta^\star, D_{t'}) + L(\theta^\star, D_t) - L(\theta^\star, D) \tag{66}$$

$$\leq \gamma \|\theta - \phi_t\|$$

$$+ \frac{1}{T} \beta \cdot \sum_{j \in [T]} W_1(P_{\phi_t}(D), P_{\phi_t}(D_j))$$

$$+ \frac{1}{T} \beta \cdot \sum_{j \in [T]} W_1(P_{\phi_{t'}}(D_{t'}), P_{\phi_{t'}}(D_j))$$

$$+ \beta \cdot W_1(P_{\theta^\star}(D), P_{\theta^\star}(D_{t'})) \tag{67}$$

We leverage Assumption 3.1 and Lemma 3.1 above. Note that the above hold for any $t \in \{j \in [T] : \phi_j \in \bar{\Theta}\}$ and $t' \in \{j \in [T] : \phi_j \notin \bar{\Theta}\}$. So we can minimize the above bound with respect to $t$ and $t'$. Moreover, noting the definition of $K(D, D_j) := \max_\theta W_1(P_\theta(D), P_\theta(D_j))$, we have (14) in the statement of the corollary. □

## C.7 Proof of Corollary 4.4

**Corollary C.4** (Corollary 4.4). *Consider the conditions and assumptions of Theorem 4.1. Then, for the pruned search space in* (15), *the optimality gap of $k$-shot HPO is bounded by*

$$\min_{t \in [T]} \left( \gamma \cdot \sqrt{\sum_{i \in [h]} \lambda_i^2} + 2\beta \cdot \max_{\theta \in \Theta} W_1(P_\theta(D), P_\theta(D_t)) \right), \tag{68}$$

*where $\lambda_i = \max \left\{ \left( \max_{j \in [T]} \phi_j[i] - \phi_t[i] \right), \left( \phi_t[i] - \min_{j \in [T]} \phi_j[i] \right) \right\}$.*

*Proof.* Let $\phi_t$ be some source-task-best HP present in the pruned HP space $\bar{\Theta}$, and let $D_t$ be the data distribution of that source task $\tau_t$ with $\phi_t = \arg\min_{\theta \in \Theta} L(\theta, D_t)$. Note that, with $\bar{\Theta}$ defined in (15), $\phi_t \in \bar{\Theta} \forall t \in [T]$.

Then we have the following by Lemma 3.1:

$$L(\hat{\theta}, D) - L(\theta^\star, D) = \underbrace{L(\hat{\theta}, D) - L(\phi_t, D)}_{(a)} + L(\phi_t, D) - L(\phi_t, D_t)$$

$$+ \underbrace{L(\phi_t, D_t) - L(\theta^\star, D_t)}_{\leq 0 \text{ by def.}} + L(\theta^\star, D_t) - L(\theta^\star, D) \tag{69}$$

$$\leq (a) + \beta \cdot W_1(P_{\phi_t}(D), P_{\phi_t}(D_t)) + \beta \cdot W_1(P_{\theta^\star}(D), P_{\theta^\star}(D_t)) \tag{70}$$

$$\leq (a) + 2\beta \cdot \max_{\theta \in \Theta} W_1(P_\theta(D), P_\theta(D_t)). \tag{71}$$

Now we need to bound the term $(a)$. We can bound this by using Assumption 3.1 and the definition of $\bar{\Theta}$ in (15) as follows:

$$(a) := L(\hat{\theta}, D) - L(\phi_t, D) \leq \gamma \|\hat{\theta} - \phi_t\| \leq \gamma \max_{\theta \in \bar{\Theta}} \|\theta - \phi_t\| = \gamma \sqrt{\sum_{i \in [h]} \lambda_i^2}, \tag{72}$$

where $\lambda_i$ is as defined in the statement of the corollary. Substituting above in (71) and minimizing over $t \in [T]$ (since the above bound holds for any $t \in [T]$) gives us (16) in the statement of the corollary. $\qquad\square$

## C.8 Proof of Corollary 4.5

**Corollary C.5** (Corollary 4.5). *Consider the conditions and assumptions of Theorem 4.1. Then, for the pruned search space in* (17)*, the optimality gap of $k$-shot HPO is bounded by*

$$\min_{t \in [T]} \left( \gamma \cdot \max_{j \in [T]} \|\phi_t - \phi_j\| + 2\beta \cdot \max_{\theta \in \Theta} W_1(P_\theta(D), P_\theta(D_t)) \right). \tag{73}$$

*Proof.* The proof of Corollary 4.5 follows along the same strategy as in the proof of Corollary 4.4, where we arrive at:

$$L(\hat{\theta}, D) - L(\theta^\star, D) \leq L(\hat{\theta}, D) - L(\phi_t, D) + 2\beta \cdot \max_{\theta \in \Theta} W_1(P_\theta(D), P_\theta(D_t)). \tag{74}$$

Then to bound the first term in the right-hand-side of the above inequality, we utilize the definition of the convex hull as follows:

$$(a) := L(\hat{\theta}, D) - L(\phi_t, D) \leq \gamma \|\hat{\theta} - \phi_t\| \tag{75}$$

$$\leq \gamma \max_{\theta \in \bar{\Theta}} \|\theta - \phi_t\| = \gamma \max_{w_1, \ldots, w_T : w_j \in [0,1], \sum_j w_j = 1} \left\| \sum_{j \in [T]} w_j \phi_j - \phi_t \right\| \tag{76}$$

$$\leq \gamma \max_{w_1, \ldots, w_T : w_t \in [0,1], \sum_t w_t = 1} \sum_{j \in [T]} w_j \cdot \|\phi_j - \phi_t\| \tag{77}$$

$$\leq \gamma \max_{j \in [T]} \|\phi_j - \phi_t\|. \tag{78}$$

Substituting above in (74) gives us (18) in the statement of the corollary. $\qquad\square$

## D Technical details for §5

### D.1 Proof of Theorem 5.1

**Theorem D.1** (Theorem 5.1). *Given source tasks $\tau_t, t \in [T]$, and their corresponding surrogate loss functions $s_t : \Theta \to \mathbb{R}$, let $\hat{\theta}$ be the result of a $k$-shot warm-started HPO with the surrogate function defined as $s(\theta) := \sum_{t \in [T]} \alpha_t(\theta) s_t(\theta)$ for a target HPO task $\tau$ with data distribution $D$. Let $\theta^\star$ be the optimal HP for the target task. Then, the optimality gap is bounded as:*

$$L(\hat{\theta}; D) - L(\theta^\star; D) \leq 2 \max_{\theta \in \Theta} \sum_{t \in [T]} \alpha_t(\theta) \left( \beta \cdot W_1 \left( P_\theta(D), P_\theta(D_t) \right) + |L(\theta, D_t) - s_t(\theta)| \right). \quad (79)$$

*Proof.* We are considering few-shot HPO with the surrogate model $s : \Theta \to \mathbb{R}$. So we assume that the HPs tried in the few-shot HPO are ones that have a lower surrogate loss values compared to the ones not tried. More precisely, if we attempted the HPs $\{\theta_i, i \in [k]\}$ in $k$-shot HPO, with the final selected HP $\hat{\theta} \in \{\theta_i, i \in [k]\}$, we know that $s(\theta) \leq s(\theta')$ for any $\theta \in \{\theta_i, i \in [k]\}, \theta' \in \Theta \setminus \{\theta_i, i \in [k]\}$ by definition.

Then we have the following bound on the optimality gap:

$$L(\hat{\theta}, D) - L(\theta^\star, D) = L(\hat{\theta}, D) - s(\hat{\theta}) + \underbrace{s(\hat{\theta}) - s(\theta^\star)}_{\leq 0 \text{ (see disc. above)}} + s(\theta^\star) - L(\theta^\star, D) \quad (80)$$

$$\leq 2 \max_{\theta \in \Theta} |L(\theta, D) - s(\theta)|. \quad (81)$$

Using the definition of $s(\theta) = \sum_{t \in [T]} \alpha_t(\theta) \cdot s_t(\theta)$ as the weighted combination of the per-source-task surrogate loss functions, we can show the following:

$$|L(\theta, D) - s(\theta)| = \left| L(\theta, D) - \sum_{t \in [T]} \alpha_t(\theta) \cdot s_t(\theta) \right| \quad (82)$$

$$= \left| L(\theta, D) - \sum_{t \in [T]} \alpha_t(\theta) \cdot L(\theta, D_t) + \sum_{t \in [T]} \alpha_t(\theta) \cdot L(\theta, D_t) - \sum_{t \in [T]} \alpha_t(\theta) \cdot s_t(\theta) \right| \quad (83)$$

$$= \left| \sum_{t \in [T]} \alpha_t(\theta) \cdot (L(\theta, D) - L(\theta, D_t)) + \sum_{t \in [T]} \alpha_t(\theta) \cdot (L(\theta, D_t) - s_t(\theta)) \right| \quad (84)$$

$$= \left| \sum_{t \in [T]} \alpha_t(\theta) \cdot ((L(\theta, D) - L(\theta, D_t)) + (L(\theta, D_t) - s_t(\theta))) \right| \quad (85)$$

$$\leq \sum_{t \in [T]} \alpha_t(\theta) \cdot (|L(\theta, D) - L(\theta, D_t)| + |L(\theta, D_t) - s_t(\theta)|), \quad (86)$$

where the last inequality leverages the fact that the weights $\alpha_t(\theta) \in [0, 1] \forall \theta \in \Theta, \forall t \in [T]$. Now we can leverage Lemma 3.1 to bound the $|L(\theta, D) - L(\theta, D_t)|$ term in (86), and then substitute this in (81) gives us the bound (19) in the statement of the theorem. □

### D.2 Proof of Corollary 5.1

**Corollary D.1** (Corollary 5.1). *Under conditions of Theorem 5.1 and Assumption 3.2, we bound the optimality gap as*

$$L(\hat{\theta}; D) - L(\theta^\star; D) \leq 2\epsilon + 2\beta \cdot \max_{\theta \in \Theta} \sum_{t \in [T]} \alpha_t(\theta) W_1 \left( P_\theta(D), P_\theta(D_t) \right). \quad (87)$$

*Proof.* Given Assumption 3.2, we have a universal bound on the $|L(\theta, D_t) - s_t(\theta)| \leq \epsilon$ term in the bound (19) of Theorem 5.1. Plugging this value in (19) and noting that $\sum_{t \in [T]} \alpha_t(\theta) = 1 \forall \theta \in \Theta$ gives us the statement of the corollary. $\qquad \square$

## D.3 Proof of Corollary 5.2

**Corollary D.2** (Corollary 5.2). *Under conditions of Theorem 5.1 and Assumptions 3.3 & 3.4, and further denoting by the set $\Phi_t = \{\theta_{t,i}, i \in [K]\}$ the HPs evaluated during the HPO of task $\tau_t$ for each $t \in [T]$, we can bound the optimality gap as*

$$L(\hat{\theta}; D) - L(\theta^\star; D) \leq 2\epsilon + 2 \cdot \max_{\theta \in \Theta} \sum_{t \in [T]} \alpha_t(\theta) \left( \beta \cdot W_1 \left( P_\theta(D), P_\theta(D_t) \right) + (\gamma + \omega) \min_{i \in [K]} \|\theta - \theta_{t,i}\| \right).$$
(88)

*Proof.* Given Assumptions 3.4 and 3.3, we can bound the term $|L(\theta, D_t) - s_t(\theta)|$ term in the bound (19) of Theorem 5.1 as follows:

$$|L(\theta, D_t) - s_t(\theta)| \leq |L(\theta, D_t) - L(\theta_{t,i}, D_t) + L(\theta_{t,i}, D_t) - s_t(\theta)| \text{ for any } i \in [K] : \theta_{t,i} \in \Phi_t \quad (89)$$

$$\leq |L(\theta, D_t) - L(\theta_{t,i}, D_t)| + |L(\theta_{t,i}, D_t) - s_t(\theta)| \quad (90)$$

$$\leq \underbrace{\gamma \cdot \|\theta - \theta_{t,i}\|}_{\text{Assumption 3.1}} + |L(\theta_{t,i}, D_t) - s_t(\theta_{t,i}) + s_t(\theta_{t,i}) - s_t(\theta)| \quad (91)$$

$$\leq \gamma \cdot \|\theta - \theta_{t,i}\| + |L(\theta_{t,i}, D_t) - s_t(\theta_{t,i})| + |s_t(\theta_{t,i}) - s_t(\theta)| \quad (92)$$

$$\leq \gamma \cdot \|\theta - \theta_{t,i}\| + \underbrace{\epsilon}_{\text{Assumption 3.4}} + \underbrace{\omega \cdot \|\theta - \theta_{t,i}\|}_{\text{Assumption 3.3}} \quad (93)$$

Since the above holds for any $i \in [K]$, so we can minimize this over all $\theta_{t,i}$, leveraging proximity to all HPs experienced during the HPO for the source task $\tau_t$. So minimizing the above with respect to $i \in [K]$ and then substituting this bound in (19) of Theorem 5.1 gives us the bound (21) in the statement of the corollary. $\qquad \square$