# OpenReview forum: "On the Optimality Gap of Warm-Started Hyperparameter Optimization"
_automl.cc/AutoML/2022/Track/Main — AutoML-Conf 2022 (Main Track)_

### Official Review · Reviewer_SRGx · 2022-04-02

**Potential Impact On The Field Of Automl Rating:** 3
**Technical Quality And Correctness Rating:** 3
**Clarity Rating:** 4

**Summary Of Contributions:**

The authors propose a theoretical framework to evaluate various transfer-learning methods that are used to "warm-start" a HPO job on a target task using HPO runs from multiple source tasks. Authors cover a variety of popular methods from the literature and provide a lower bound for all of those. Apart from providing mathematical bounds, authors also provide intuitions regarding the bounds and use it to further validate certain empirical observations from the literature. The paper is entirely theoretical and does not have any empirical component as the work analyzes existing techniques which already provide experimental results.

**Clarity:**

Authors perform a good job in presenting the materials in a cohesive way to a reviewer without a strong theoretical background. The flow of the information and the narrative seems natural and all assumptions are clearly listed down before discussing each lemma or corollaries.

**Ethics Details (Optional):**

This paper is purely theoretical and therefore no ethics concern as there are no experiments done or new datasets or benchmarks proposed.

**Overall Review:**

Strength
------------
* The paper proposes a comprehensive theoretical framework to analyze various HPO warm-starting techniques by analyzing their lower bound. The framework can accommodate both pruning based methods and surrogate function transfer methods.
* Paper is nicely written and all the assumptions and limitations about each theorem and lemma are clear from the paper. Whenever certain assumptions are stronger that what is encountered in real life, authors mention that and also provide bounds for alternative (weaker) assumptions which are more realistic.
* The bounds obtained by the authors intuitively make sense to me. They also seem to be correct for a couple of proofs that I have checked in the appendix. The fact that the results obtained from the theoretical analysis helped to verify empirical observations from the literature (convex-hull vs bounding boxes) further solidify the work.

Weakness
------------
* I was expecting the authors to provide some novel insight or algorithm based on all the analysis they have done. The results do not provide additional help to the practitioners between choosing one strategy over the other because to me, it looks like the largest contributing factor for all methods is the domain-gap between source and target tasks which is a hard problem itself.
* On the same line, authors use the 1-wasserstein metric between datasets to measure domain-gap. They also mention that in practice, we can use dataset meta-features to approximate this distance. I'd have to liked to see a discussion as to how well meta-features approximate the dataset distance metric used in this paper.

**Potential Impact On The Field Of Automl:**

From a theoretical standpoint, this paper can have a strong impact in this specific sub-area (transfer-learning/warm-starting for HPO) of AutoML as new methods can use the framework proposed in the paper to compare the lower bounds obtained by their method with respect to the existing ones apart from only validating their hypothesis with empirical results.

From a practical standpoint, this work will have limited impact because i) the authors do not propose any new method/insight using their theoretical observation and ii) most of the hypotheses proved in the paper seems intuitive e.g. HPO transfer performance is heavily dependent on the source-target task domain difference.

**Reproducibility:**

This paper is purely theoretical and therefore reproducibility is not a concern.

**Review Confidence:**

3: You are fairly confident in your assessment. It is possible that you did not understand some parts of the submission or that you are unfamiliar with some pieces of related work.

**Review Rating:**

5: Accept, good paper

**Review Summary:**

The authors have developed a holistic theoretical framework to evaluate various warm-starting strategies for HPO. The framework is comprehensive, intuitive, easy to understand and extendable to future work. On the other hand, the practicality of this analysis is slightly limited. Overall, I found the paper to be an important work for both analyzing past work and making future work on this topic more theoretically grounded and therefore, recommend accept.

**Technical Quality And Correctness:**

* The proposed lemmas and bounds described in this paper seem correct to me.
* The assumptions used to prove the bounds seem reasonable to the most extent.
* The authors propose a general framework and use it to cover two styles of HPO-transfer-learning e.g. search-space pruning and transferring the surrogate function itself. The framework is sufficient to prove the optimality bounds in both cases.
* The bounds also theoretically justify empirical results of a few techniques from the literature, thereby further showing the correctness of the claims.

---

### Official Review · Reviewer_xs8s · 2022-04-02

**Potential Impact On The Field Of Automl:** N/A for reproducibility reviewers
**Potential Impact On The Field Of Automl Rating:** 4
**Technical Quality And Correctness:** N/A for reproducibility reviewers
**Technical Quality And Correctness Rating:** 4
**Clarity:** N/A for reproducibility reviewers
**Clarity Rating:** 4

**Summary Of Contributions:**

N/A for reproducibility reviewers

**Overall Review:**

The authors only performed a theoretical analysis.

**Reproducibility:**

The authors only performed a theoretical analysis.

**Review Confidence:**

3: You are fairly confident in your assessment. It is possible that you did not understand some parts of the submission or that you are unfamiliar with some pieces of related work.

**Review Rating:**

6: Strong accept, should be highlighted

**Review Summary:**

N/A for reproducibility reviewers

---

### Official Review · Reviewer_N1dt · 2022-04-03

**Potential Impact On The Field Of Automl Rating:** 2
**Technical Quality And Correctness Rating:** 3
**Clarity Rating:** 3

**Summary Of Contributions:**

The authors provide theoretical analysis of the optimality gap of the hyperparameters obtained via warm-started few-shot HPO. Using their theoretical analysis they identify situations where transfer surrogate loss functions perform better than HP space pruning.

**Clarity:**

It should be described more clearly how the theoretical analysis gives guidance to identify when some schemes are better. After reading the paper it is not too clear to me how I could actually use those theoretical results. More detailed examples could be provided. Also transfer surrogate loss functions and HP space pruning methods could be described with more example approaches from literature. It is not particularly clear what proportion of meta-learning/HPO methods this actually encapsulates and hence how much impact the work has. To further improve the clarity, it would be better to avoid very long sentences that span e.g. 3 or 4 lines. On the other hand, I like how authors include a section to define the preliminaries - this certainly improves the clarity and makes it easier to follow the proofs.

There are a few minor typos/grammar mistakes, e.g.:
L102: However, the advantage of the meta-features rely heavily
L126: a output


**Overall Review:**

Positive aspects:
* The paper provides a theoretical analysis that can be useful to decide when to prefer certain families of HPO methods.
* The analysis is detailed, assumptions are clearly described and the theoretical analysis is properly written up.
* The authors provide a detailed technical analysis of limitations of their their work.

Negative aspects:
* Currently it is difficult to identify what are the implications of the theoretical analysis and how it can be useful to make decisions on what HPO or meta-learning methods to use.
* The authors say that their theoretical results give the guidance (e.g. in abstract, introduction or conclusion), but it is either so technical that it is hard to grasp or the details of such guidance are actually missing.
* It is also not clear to how many HPO or meta-learning methods the results apply - is it some very specialized family which would imply a rather small impact of the analysis or is it more general? * Generally I would appreciate more examples for those approaches and some discussion.

Overall I think it is a solid work with some interesting results, but the current presentation makes it difficult to understand how to use the results of the analysis.


**Potential Impact On The Field Of Automl:**

The findings are generally interesting for the AutoML community as it gives us deeper insights into the theory behind the selected families of meta-learning approaches. However, I find it quite difficult to see the implications of the work after reading the paper. The work may have potential for more impact if additional interpretations are provided and it is linked more precisely to the empirical observations in literature - with some examples of how it relates to different specific approaches and what it says about them.

**Reproducibility:**

It is a theoretical work and the proofs seem to be sufficiently detailed.

**Review Confidence:**

2: You are willing to defend your assessment, but it is quite likely that you did not understand the central parts of the submission or that you are unfamiliar with some pieces of related work.

**Review Rating:**

5: Accept, good paper

**Review Summary:**

I currently select marginally above the acceptance threshold, but the paper can move quite easily to “Accept, good paper” if the paper gives better guidance on the practical implications of the theoretical results. Clarifications to the issues described in various sections of the review would be appreciated within the rebuttal. The paper would benefit a lot from connecting the theoretical analysis to practical guidance to a larger extent.

**Technical Quality And Correctness:**

The work is of high quality, the authors make it clear what the assumptions are and the proofs appear to be correct after quickly reading them. Perhaps one detail that I find slightly worrying regarding technical quality: I get the impression that the statements in the abstract, introduction and conclusion about the general implications are somewhat exaggerated since from the theoretical analysis it is not very clear how it gives guidance when e.g. one scheme is better than another.

---

### Official Review · Reviewer_bVnP · 2022-04-04

**Potential Impact On The Field Of Automl Rating:** 3
**Technical Quality And Correctness Rating:** 3
**Clarity Rating:** 3

**Summary Of Contributions:**

The paper presents a theoretical study of the optimality gap of warm-started hyperparameter optimization algorithms, for a variety of warm-starting strategies that either restrict the search space or use previously fitted surrogate functions.
The authors discuss the implications of such bounds concluding that the optimality gap of surrogate function transfer is smaller than that achievable by studied pruning strategies when using adaptive weights.
In particular, this shows the importance of the weighting strategies in these second classes of methods.
The paper does not include any numerical experiments.

**Clarity:**

The paper is overall clear, but some passages would benefit from further proofreading (e.g. sec 2 is full of repetitions).
The mathematical notation is a bit involved but probably this is a necessity. There are some symbols that are not properly introduced, such as the tilde{O} in equation (2) and onward (big O with log factors??).

Minor:
- In the appendix, rewriting the theorems' statement would increase readability
- It is not clear to me how the "local search" is supposed to be performed in section 4.


**Overall Review:**

Strengths:
- Interesting theoretical analysis of many warm-starting HPO strategies
- Clear and interesting discussion of the obtained bounds
- Clear exposition of the problem setting, including motivation.

Weaknesses:
- Many assumptions that may limit the applicability of the results in practice
- No empirical validation/visualization of the bounds
- The work would benefit from an additional round of proofreading

**Potential Impact On The Field Of Automl:**

Theoretical analysis of HPO methods is lagging behind empirical advances, and this paper somewhat tries to close the gap by focusing on few-shot warm-started HPO algorithms.
There are a series of limitations to the analysis the authors provide, most of which are clearly discussed in the paper.
Additionally, I would say that a further limitation is that the setting consider is quite homogeneous, meaning that it is assumed that all the tasks feature the same loss and that it may sound quite optimistic to assume that for the source tasks optimal hyperparameters have been found.
Also, the paper seems not to focus on a specific HPO technique (other than putting a certain emphasis on model-based algorithms) and rather fully commits to the few-shot, many-source-tasks setting.
Nevertheless, I believe the results reported, including the type of approach that the authors follow, can have a certain impact in the field and stimulate further analysis and empirical investigation.

**Reproducibility:**

N/A

**Review Confidence:**

3: You are fairly confident in your assessment. It is possible that you did not understand some parts of the submission or that you are unfamiliar with some pieces of related work.

**Review Rating:**

5: Accept, good paper

**Review Summary:**

The results contained in the paper may be of interest for the HPO community, revitalizing interest in few-shot HPO techniques and theoretical analysis of HPO algorithms.
The limitations of the analysis are overall understandable and can be addressed by future work.

----------------------------------
After rebuttal
----------------------------------
I thank the authors for their reply and confirm my recommendation to accept the paper.

**Technical Quality And Correctness:**

Superficially the results seem correct, although I have not checked all the details of all the proofs.

---

### Official Review · Reviewer_ydGB · 2022-04-05

**Potential Impact On The Field Of Automl Rating:** 1
**Technical Quality And Correctness Rating:** 4
**Clarity Rating:** 3

**Summary Of Contributions:**

The authors present a general framework for the analysis of few-shot HPO, with a focus on the optimallty gap across varying warm-starting techniques in HPO.

**Clarity:**

For the most part, the work is clear. I refer the authors to line 95, to fix the beginning of the sentence

**Overall Review:**

Check Review Summary.

**Potential Impact On The Field Of Automl:**

The paper provides an interesting study of the optimality gap in light of transfer learning solutions for HPO. Nevertheless, the paper has potentially very minimal impact on the field of AutoML.

**Reproducibility:**

Not reproducible due to the theoretical nature of the contribution.

**Review Confidence:**

4: You are confident in your assessment, but not absolutely certain. It is unlikely, but not impossible, that you did not understand some parts of the submission or that you are unfamiliar with some pieces of related work.

**Review Rating:**

2: Reject, not good enough

**Review Summary:**

The authors present a general framework to analyze the optimality gap in few-shot HPO. The idea extends in the transfer learning context of HPO where sequential model-based optimization has already been applied to a collection of source tasks.

**Strengths**:

- The paper addresses an interesting aspect of HPO that has been overlooked by the community.

- Sound derivations

- The authors look at several ways by which transfer learning has been employed to improve HPO, and derive several respective theoretical guarantees.

**Weaknesses**: The theoretical guarantees that define the approximation error to the optimal hyperparameter, i.e the optimality gap are governed by several assumptions, including surrogates being Lipschitz continuous, and low approximation error, more on that below.

- Lemma 3.1 defines the 1-Wasserstein distance between two distributions. Distribution P_\theta(D) is not clearly defined. Furthermore, it would differ depending on the sample size, and this point is not further discussed. There is no clear motivation as to why Wasserstein distance is used.

- Going through the derivations, I was surprised to see that meta-features are not at all used as a proxy to measure data set similarity, as was clearly stated in the related work sections. The authors cover this topic, even though they miss a prominent paper on learning meta-features [1], and their applications in few-shot HPO [2].

- Assumption 3.3/3.4 Although restrictive, these assumptions overlook a major aspect of any surrogate function, namely modeling uncertainty. It is not mentioned throughout the derivations, and one would assume, from the current literature trends, that uncertainty would play an important role in deriving the optimality gap.

- It would also be interesting to look at the relative performance of HPs, in measuring the quality of the surrogate, instead of the approximation error.

References:

[1] Jomaa, Hadi S., Lars Schmidt-Thieme, and Josif Grabocka. "Dataset2vec: Learning dataset meta-features." Data Mining and Knowledge Discovery 35.3 (2021): 964-985.

[2] Jomaa, Hadi Samer, et al. "Transfer Learning for Bayesian HPO with End-to-End Landmark Meta-Features." Fifth Workshop on Meta-Learning at the Conference on Neural Information Processing Systems. 2021.

**Technical Quality And Correctness:**

The derivations and assumptions presented in the main paper appear to be technically correct.

---

### Meta-Review · Area_Chair_Jocr · 2022-05-05

**Recommendation:** Accept
**Confidence:** 4

**Metareview:**

The paper presents theoretical bounds on performance of transfer learning methods.
Various methods recently proposed in the literature are analysed with a novel framework which will be helpful to improve foundational aspects of transfer learning HPO methods.

All reviewers acknowledged the novelty of the paper and potential impact, one of the biggest concern was the amount of insights coming out of the paper, but the authors clarified this aspect (two raised their scores). The authors also answered the pointed raised by the reviewers and all but one reviewers argue for accepting the paper (except one reviewer who did not acknowledge the author response).

I recommend accepting the paper which will be a valuable contribution to improve theoretical guarantees of transfer learning methods for HPO.

---

### Decision · Program_Chairs · 2022-05-13

Accept